# Mathematical and statistical approaches in epidemiological investigation of hospital infection: A case study of the 2015 Middle East Respiratory Syndrome outbreak in Korea

Youngsuk Ko[1], Eunok Jung[2]*

**1** Institute of Mathematical Sciences, Konkuk University, Seoul, Republic of Korea, **2** Department of Mathematics, Konkuk University, Seoul, Republic of Korea

* junge@konkuk.ac.kr

**Data Availability Statement:** The data that support the findings of this study are openly available in

## Abstract

Mathematical and statistical methods are invaluable in epidemiological investigations, enhancing our understanding of disease transmission dynamics and informing effective control measures. In this study, we presented a method to estimate transmissibility using patient-level data, with application to the 2015 MERS outbreak at Pyeongtaek St. Mary's Hospital, the Republic of Korea. We developed a stochastic model based on individual case data to derive a likelihood function for disease transmission. Through scenario-based analysis, we explored transmission dynamics, including the role of superspreaders, and investigated how mask-wearing impacted infection control within the hospital. Our findings indicated that the superspreader during the MERS outbreak had approximately 25 times higher transmissibility compared to other patients. Under scenarios of prolonged hospital transmission periods, the number of cases could potentially increase threefold. The impact of mask-wearing in the hospital was significant, with reductions in the epidemic scale ranging from 17% to 77%, depending on the type of mask and intervention intensity. This study quantifies key risk factors in hospital-based transmission, demonstrating the effectiveness of intervention measures. The methodology developed here can be readily adapted to other infectious diseases, providing valuable insights for future outbreak preparedness and response strategies.

## Introduction

An epidemiological investigation is a systematic method used to determine the cause, source, and spread of a disease within a population, usually following an unusual increase in the number of cases [1, 2]. This process typically involves steps like verifying the outbreak, defining and identifying additional cases, using epidemiological methods to understand the disease transmission dynamics, and collecting data pertaining to cases, their characteristics, and potential risk factors [2, 3].These data are directly utilized to estimate model parameters and

figshare at https://doi.org/10.6084/m9.figshare.25981987.v1.

**Funding:** This paper was supported by Konkuk University in 2022 in the form of funding awarded to EJ.

**Competing interests:** The authors declare no conflicts of interest.

simulate realistic outbreak patterns. This allows for evidence-based implementation of control measures to prevent further spread of the disease and reduce the risk of future outbreaks [3, 4].

Mathematical and statistical approaches enhance the significance of epidemiological investigations. Information aggregated through these investigations can be processed to understand the unique characteristics of specific infectious diseases, such as their incubation or latent periods, and presented as statistical distributions to aid future research [5–7]. For instance, during the COVID-19 pandemic in South Korea, detailed epidemiological information on individual cases—including estimated exposure dates, symptom onset dates, and reporting dates—was collected and analyzed using mathematical and statistical methods until the Omicron variant became dominant [8–11]. These analyses revealed changes in contact patterns between age groups during COVID-19 pandemic, which differed from those observed before the pandemic. This information was then used for short- and long-term predictions, considering factors such as vaccine prioritisation and policy decisions. By integrating epidemiological investigations with mathematical and statistical analyses, this approach provides evidence that enhances the reliability and accuracy of model results [8–13].

In this study, we introduce a process for interpreting epidemiological investigations using mathematical and statistical methods to estimate key parameters and transmission rates. As a case study, we apply this method to an outbreak at Pyeongtaek St. Mary's Hospital (PMH), the initial site of hospital transmission during the 2015 Middle East Respiratory Syndrome (MERS) outbreak in Korea, where 36 out of 186 total cases were identified [14–16]. Studies conducted in endemic regions described the transmission pathways originating from environmental factors, progressing to human hosts, and ultimately reaching healthcare facilities [17–19]. Our research focused on non-endemic regions where such intermediate host transmission may not play a significant role. To investigate transmission rates in detail, we categorized individuals into superspreaders, healthcare workers (HCWs), patients, and visitors based on actual epidemiological investigations. The spread in Korea was largely due to superspreaders, patients who caused secondary infections in more than six people [20–23]. Of the 186 cases, it was suspected that 15 transmitted the disease to others, with five identified as superspreaders.

Simultaneously with the estimation of the transmission rate, we developed a model that reflects these investigations and simulates a realistic outbreak. A scenario-based analysis was introduced to quantify risk factors such as the infectious period of infected individuals, including superspreaders, and the effectiveness of mask mandates within the hospital. The methodology employed in this study transforms the dynamics of hospital infections into a likelihood function, allowing for its application to other hospitals or infectious disease outbreaks. By focusing on the estimation of transmission rates among heterogeneous hosts in certain spaces like hospitals, we expect these findings to significantly enhance both reactive and preventive measures. This approach, which has been successfully utilized in various contexts, including human-to-human and animal-to-animal infectious diseases, underscores the potential for improving infection control strategies [8–11, 24, 25].

## Materials and methods

### Probabilities and likelihoods during the outbreak

Two necessary pieces of information expected from the epidemiological investigation are: (1) when were the infectious individuals suspected of transmitting the disease and (2) when were the infected individuals infected with the disease (or the time range in which they were exposed to the disease). These factors influence the probability of individuals becoming infected at certain time points. Let us consider a classic SIR model formulated using ordinary differential equations for a susceptible population [26]. A susceptible host can be infected by

an infectious host at a rate $\beta$. Let N indicate the total number of hosts. When assuming frequency-dependent transmission, ignoring natural birth and death [27], and fixing the number of infectious hosts as $N_I$ (estimated based on the first piece of information gathered from the epidemiological investigation mentioned above), the following equation is obtained:

$$\frac{dS}{dt} = -\beta S \frac{N_I}{N}.$$ (1)

This equation transitions to a linear form, facilitating a straightforward resolution. Given the assumption that the initial S is one to consider the status of a single host and N is a constant, the equation is solved as follows:

$$S(t) = \exp\left(-\beta S \frac{N_I}{N} t\right).$$ (2)

This solution represents the probability that the host remains in the susceptible state for up to time t. When considering a unit of time, defined as $t = 1$, the subsequent probability is conceptualized as follows [28–30]:

$$q_s = \exp\left(-\beta S \frac{N_I}{N}\right).$$ (3)

In this framework, $q_S$ is interpreted as the probability of a host maintaining an uninfected status over a unit of time. Conversely, the probability of a host becoming infected is represented as $q_I$, and expressed as follows:

$$q_I = 1 - q_s = 1 - \exp\left(-\beta S \frac{N_I}{N}\right).$$ (4)

Let us consider a situation in which multiple hosts coexist, as shown in Fig 1. At time t, there are N hosts in total, with $N_S$ susceptible and $N_I$ infectious hosts. After a unit of time $(t+1)$, if $X$ hosts are infected and $Y$ hosts are not infected $(X+Y=N)$, using previously

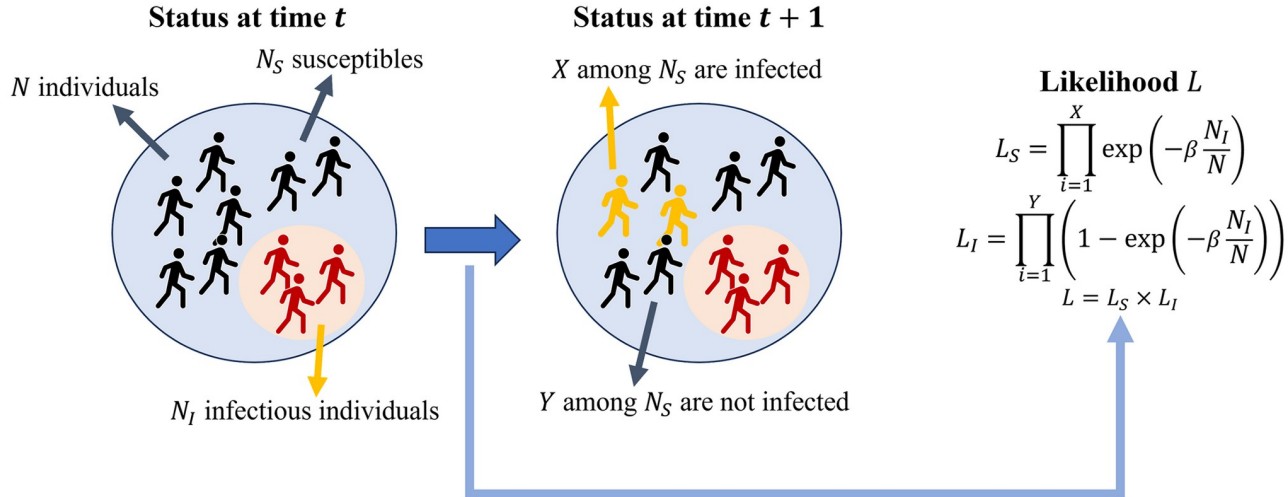

**Fig 1. Derivation of the likelihood function for a unit timestep considering both unsuccessful and successful disease transmissions.**

calculated probabilities, we can calculate the following likelihood function ($L$):

$$L = \prod_{i=1}^{X} \exp\left(-\beta S \frac{N_I}{N}\right) \times \prod_{i=1}^{Y} \left(1 - \exp\left(-\beta S \frac{N_I}{N}\right)\right). \tag{5}$$

Since the epidemiological investigation is based on actual incidences, the value of $\beta$ that maximizes this likelihood function can represent the reality. Therefore, our goal is to determine the value of $\beta$ that maximizes $L$. In this study, the Metropolis-Hastings algorithm was used to sample the parameters to find the value of $\beta$ [31]. This explanation describes progress in one unit of time, whereas the actual case application considers multiple unit time steps and is expressed as the product of all probabilities. To apply this method to real data, the exact time range of exposure for each host is required, which is the second piece of information mentioned at the beginning of this subsection.

## Application of maximum likelihood estimation (MLE) to MERS nosocomial spread in Korea and model formulation

Infectious hosts transmit the disease through viral shedding. We assumed that there is heterogeneity in the successful transmission based on contact frequency, duration, and distance between the infectee and infector, considering the types of hosts (HCWs, patients, and visitors) within the hospital setting. Note that this study only considers human-to-human transmission, as the Republic of Korea is not located in an endemic region. Kim provided insights into individual hosts considering their types, encompassing their anticipated exposure times, expected transmission times, isolation times, and the number of individuals present in PMH during the 2015 MERS outbreak in Korea [14]. Let $\beta_{AB}$ denote the transmission rate, where the subscripts $A$ and $B$ indicate the infector and infectee types, respectively. Symbols $\Omega$, $H$, $P$, and $V$ as subscripts indicating the types of hosts as superspreaders, HCWs, patients, and hospital visitors, respectively. We classified individuals as either infected or uninfected. The groups of infected and uninfected individuals are represented as $\Lambda_I$, and $\Lambda_S$, respectively. Let $D_i$ and $\tilde{D}_i$ be the identifiers for the types of hosts in $\Lambda_S$ and $\Lambda_I$, respectively, and $T$ be the discrete time points. Considering the number of infectors at a specific time $k$ is $\mathbf{I}_j(k)$, where the subscript $j$ indicates the host type, the likelihood of hosts in $\Lambda_S$ and $\Lambda_I$ is derived as follows:

$$
\begin{aligned}
L_S &= \prod_{i \in \Lambda_S}\left\{\prod_{k \in T} \exp\left(\sum_{j \in \{\Omega, H, P, V\}} -\beta_{jD_i} \frac{\mathbf{I}_j(k)}{N}\right)\right\}, \\
L_I &= \prod_{i \in \Lambda_I}\left\{\prod_{k \in T}\left(1 - \exp\left(\sum_{j \in \{\Omega, H, P, V\}} -\beta_{j\tilde{D}_i} \frac{\mathbf{I}_j(k)}{N}\right)\right)\right\}, \\
L(\mathrm{B}) &= L_S \times L_I, \\
\text{where } \mathrm{B} &= \{\beta_{HH}, \beta_{HP}, \beta_{HV}, \beta_{PH}, \beta_{PP}, \beta_{PV}, \beta_{VH}, \beta_{VP}, \beta_{VV}, \beta_{\Omega H}, \beta_{\Omega P}, \beta_{\Omega V}\}.
\end{aligned}
\tag{6}
$$

Our goal is to find B that maximises $L(\mathrm{B})$. Note that $\beta_{HH}$, $\beta_{HP}$, and $\beta_{HV}$ are set to zero because the epidemiological investigation showed that there was no contagious period involving HCWs in PMH. In addition, symmetric transmission between patients and visitors was assumed, i.e., $\beta_{PV} = \beta_{VP}$.

In this study, we developed a susceptible-exposed-infectious-recovered (SEIR)-type model to investigate nosocomial spread, and Fig 2 provides a visual representation of our model. There were five stages of disease progression: susceptible ($S$), exposed ($E$), infectious ($I$),

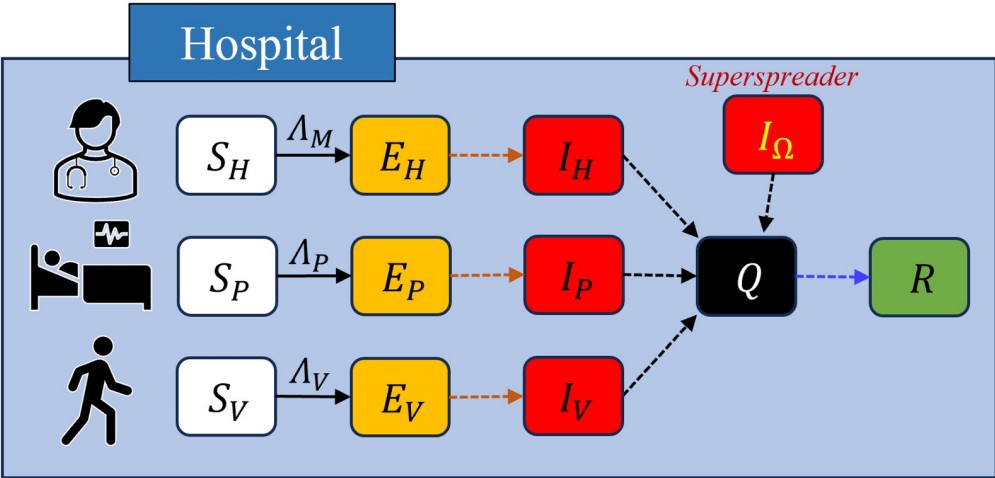

$$\Lambda_H = \frac{\beta_{HH}I_H + \beta_{PH}I_P + \beta_{VH}I_V + \beta_{\Omega H}I_\Omega}{N}$$

$$\Lambda_P = \frac{\beta_{HP}I_H + \beta_{PP}I_P + \beta_{VP}I_V + \beta_{\Omega P}I_\Omega}{N}$$

$$\Lambda_V = \frac{\beta_{HV}I_H + \beta_{PV}I_P + \beta_{VV}I_V + \beta_{\Omega V}I_\Omega}{N}$$

**Fig 2. Flow diagram of the MERS intra-hospital transmission model.**

isolated ($Q$), and recovered ($R$). The subscripts indicate the types of hosts introduced. We categorized the events that could occur during an outbreak into delayed (non-Markovian) and nondelayed (Markovian). The event where a susceptible host is exposed to infection is assumed to be Markovian since it does not involve past points in time, while all other events are assumed to be non-Markovian as they are strongly influenced by past points in time. For example, the symptom onset after exposure is significantly affected by the past exposure point. Fig 2 illustrates the different types of reactions: the non-Markovian process (transition from disease exposure, dashed line with an arrow) and the Markovian process (solid line with an arrow). To incorporate the infectious period of the hosts into the model simulation, we fitted the data assuming a gamma distribution for the samples using the built-in MATLAB function *fitdist* [32, 33]. The distribution of the delay from disease exposure to symptom onset was estimated with shape and scale parameters of 4.45 and 1.57 respectively, resulting in a mean of 6.99. For the delay from symptom onset to isolation, the estimated shape and scale parameters were 2.24 and 2.47 respectively, with a mean of 3.70. We assumed a fixed delay of 14 days from isolation to recovery [34].

By applying the estimated parameters and distributions, simulations were conducted using the modified Gillespie algorithm with 10,000 simulation runs for each scenario setting [35, 36]. To reflect the actual events in Korea, the superspreader was designated to stay in PMH for 3 days (May 15–17, 2015). The initial numbers of susceptible HCWs, patients, and visitors were 241, 263, and 389, respectively [14]. The values in Table 1 represent the population distribution for the infectious period [33]. However, due to factors such as isolation measures and ward closures in the hospital, the infectious period was adjusted by 25% (75% reduction) for

**Table 1. Description of reactions in the MERS intra-hospital transmission model.**

| Event | Reaction | dest | Reference |
|---|---|---|---|
| Infection of HCW | Markovian | Propensity: $S_H \dfrac{\beta_{HH}I_H + \beta_{PH}I_P + \beta_{VH}I_V + \beta_{\Omega H}I_\Omega}{N}$ | Fitted |
| Infection of patient | Markovian | Propensity: $S_P \dfrac{\beta_{HP}I_H + \beta_{PP}I_P + \beta_{VP}I_V + \beta_{\Omega P}I_\Omega}{N}$ | Fitted |
| Infection of visitor | Markovian | Propensity: $S_V \dfrac{\beta_{HV}I_H + \beta_{PV}I_P + \beta_{VV}I_V + \beta_{\Omega V}I_\Omega}{N}$ | Fitted |
| Symptom onset | Non-Markovian | Delay: gamma distribution, Mean: 6.99 Mean: 3.31 | [33] |
| Isolation | Non-Markovian | Delay: gamma distribution, Mean: 5.53 Mean: 3.70 | [33] |
| Recovery | Non-Markovian | Delay: fixed as 14 | [34] |

use in the baseline scenario simulation [14, 34]. This adjustment ensured that the mean number of confirmed cases from the simulations matched the actual number of cases. The results for the adjustment ratio are introduced in the following section.

Since the onset of the Coronavirus disease pandemic, the wearing of masks in health facilities has been legally enforced until April 2024 [37]. To quantify the effect of mask-wearing on preventing nosocomial spread, we conducted an additional scenario-based analysis. Parameters that had a significant impact on nosocomial spread were considered, and the following three scenarios were analysed:

- Analysis of the infectious period of hosts in the hospital: We adjusted the infectious period of the total population from 0% to 95% in our baseline scenario, which had already reduced it by 75%. The infectious period of the superspreader was fixed at 3 days.

- Analysis of adjusting the infectious period of a superspreader: We considered the infectious period for non-superspreaders as the baseline and adjusted the infectious period of the superspreader from 1 to 5 days.

- Analysis of mask-wearing interventions: We evaluated the impact of the mask mandate in hospitals. The effect varied according to the type of mask (76% for N95 and 30% for medical/surgical masks) and the level of enforcement (full effect for mandatory, half for recommended) [38]. The preventive effect of wearing a mask is reflected in a reduction in the force of infection in susceptible hosts.

## Results

### Estimation of transmission rates

The parameters sampled using the Metropolis-Hastings algorithm are represented in the form of box-whisker plots in Fig 3. Fig 3(A) and 3(B) show the transmission rates of non-super-spreaders and superspreaders, respectively. The mean values of sampled transmission rates $\beta_{PH}$, $\beta_{PP}$, $\beta_{PV}$, $\beta_{VH}$, and $\beta_{VV}$ were 0.04, 0.61, 0.01, 0.17, 0.01, and 0.05 (95% credible interval (CrI) [0.00, 0.12], [0.42, 0.84], [0.00, 0.05], [0.01, 0.49], and [0.00, 0.16]), respectively, among which the transmission rate between patients ($\beta_{PP}$) was estimated to be the highest. The mean superspreader-induced transmission rates $\beta_{\Omega H}$, $\beta_{\Omega P}$, and $\beta_{\Omega V}$ were estimated to be 4.27, 15.04,

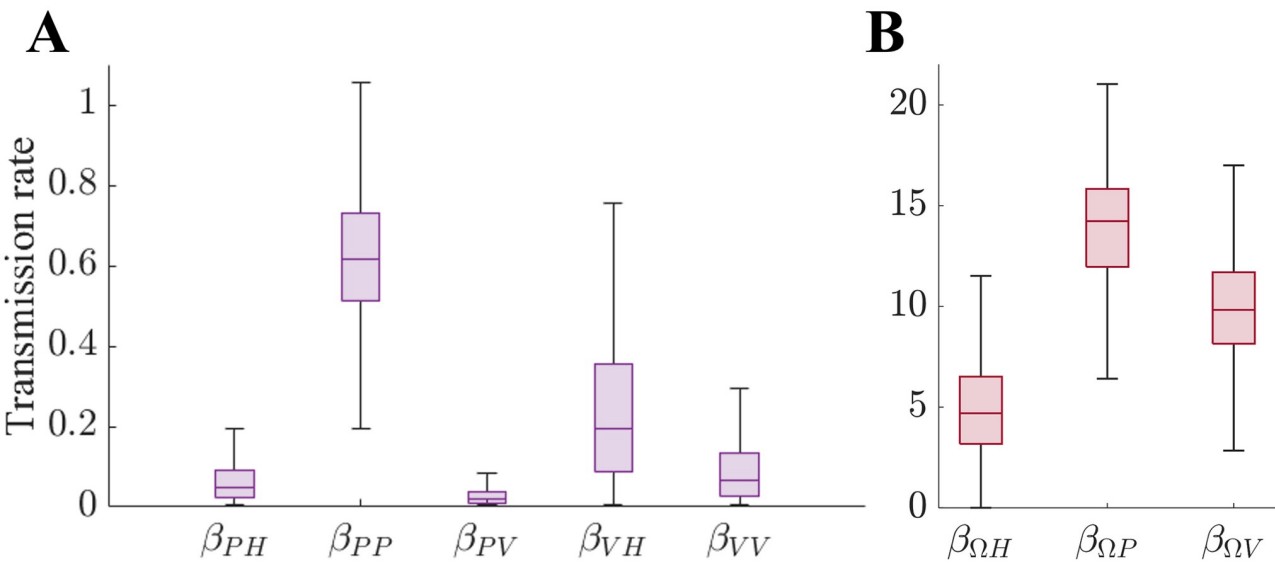

**Fig 3. Distribution of sampled transmission rates in Pyeongtaek St. Mary's Hospital during the 2015 MERS outbreak in Korea.** (A) Transmission rates induced by non-superspreaders, (B) transmission rates induced by a superspreader.

and 10.57 (95% CrI [1.69, 7.97], [9.89, 21.19], and [7.09, 14.82]). The superspreader-induced transmission rate among patients was estimated to be the highest.

## Simulation of the baseline scenario

The simulation results for the baseline scenario are shown in Fig 4. Fig 4(A) shows the cumulative number of confirmed cases over time. The grey area in the graph represents the 95% confidence interval (CI), and the dark curve indicates the mean of the simulation runs. Fig 4(B) shows the distribution of confirmed cases. The mean number of confirmed cases was 36.12, which was close to the actual number of cases (36), and the 95% CI was from 23 to 50. The state variables $E$, $I$, and $Q$ are visualised in Fig 4(C), and the prevalence (proportion of $E$, $I$, and $Q$ in the total population) is represented in Fig 4(D). Based on the mean, we observed that $E$, $I$, and $Q$ reached 28.62, 4.73, and 31.42 (3.3, 8.1, and 18.2 days after primary case onset). The possible peak in the 95% CI prevalence was 6%.

## Scenario-based study

To conduct the baseline scenario simulation, we reduced the infectious period by 75% compared to the population distribution, which includes the suspected infectious period outside the hospital, such that the mean value of the simulation runs follows the actual number of cases. Fig 5 shows the mean and 95% CI of the confirmed cases, indicated by reduction varying from 0% to 95%. The cyan dotted vertical line indicates the value used in the baseline scenario, and the magenta horizontal dotted line represents the actual number of confirmed cases. The mean (95% CI) number of confirmed cases changed from 100.40 ([39, 140]) to 30.70 ([22, 47]) as the reduction factor varies from 0% to 95%.

In the baseline scenario, the infectious period of the superspreader was fixed at 3 days. To observe the impact on the scale of the outbreak when this period varies between 1 and 5 days, Fig 6 shows the distribution of confirmed cases according to the infectious period of the

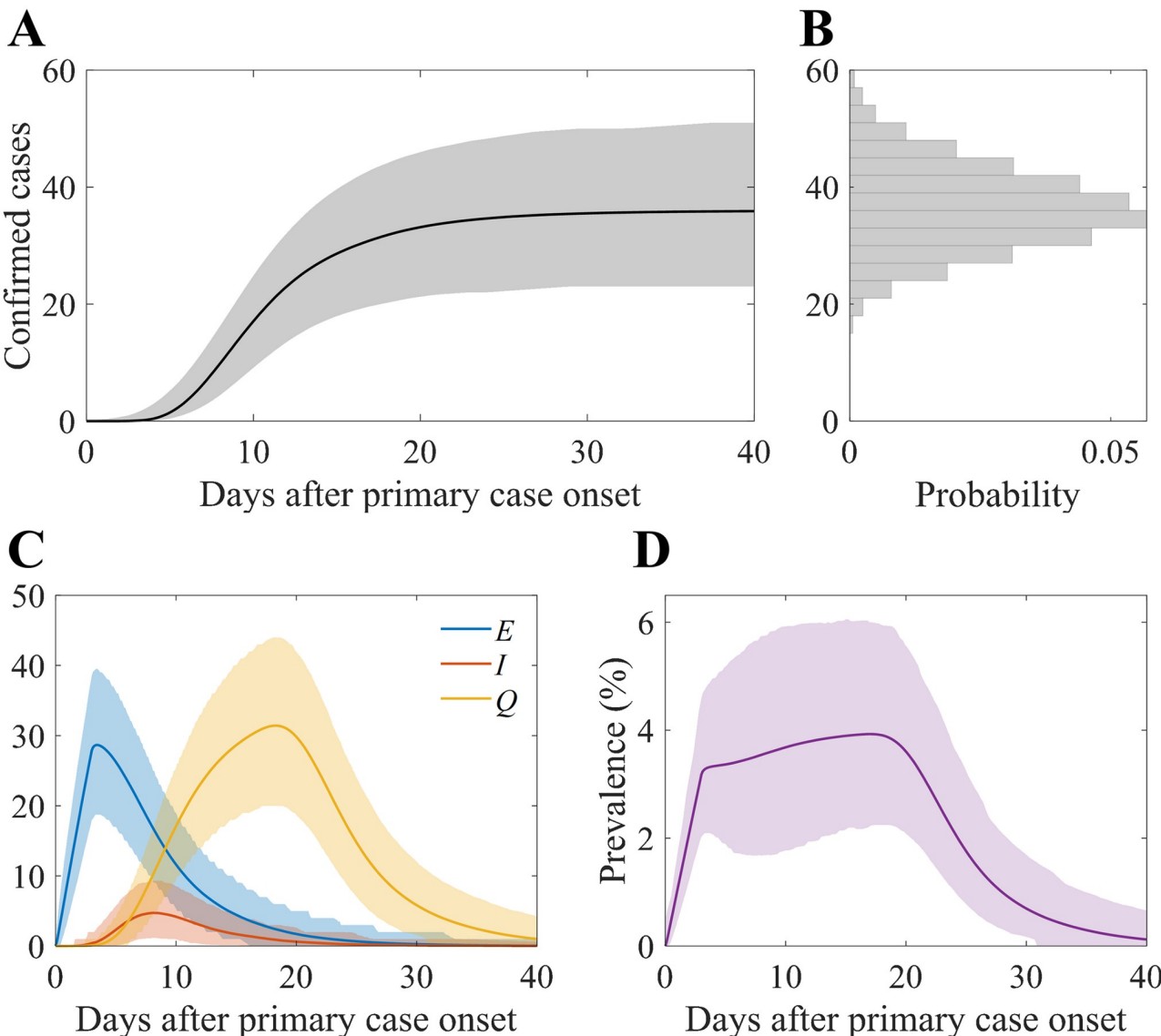

**Fig 4. Baseline simulation results of the number of confirmed cases.** (A) Cumulative confirmed cases over time, (B) distribution of the confirmed cases from simulation runs, (C) number of exposed, infectious, and isolated hosts over time, and (D) prevalence (proportion of exposed, infectious, and isolated hosts) over time. In panels A, B, and C, solid curves indicate mean value and coloured areas indicate 95% CI.

superspreader. The number of confirmed cases ranged from a minimum mean of 13.01 (95% CI [6, 23]) to a maximum of 57.84 (95% CI [41, 76]).

## Preventive effect of mask mandates

The results of the simulations considering the mask mandates are visualised in Fig 7. Fig 7(A) indicates the type of mask worn by HCWs, patients, and visitors for each detailed scenario. Fig 7(B) presents the simulation results considering mask mandates for each detailed scenario as an odds ratio of the number of confirmed cases compared with the baseline scenario, where mask mandates are not applied. The mean effect varied depending on the type of mask (N95 and medical or surgical masks), at 76% and 30%, respectively [38].

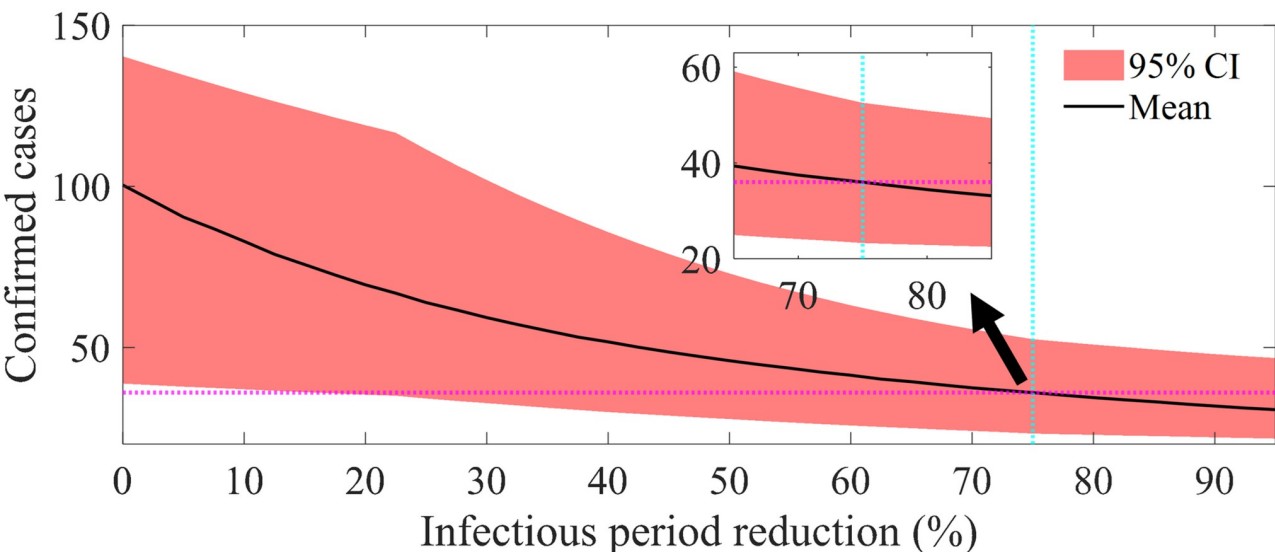

**Fig 5. Number of confirmed cases varying with change in the infectious period reduction.** Dark curve indicates simulation mean and red area covers 95% CI.

The highest reduction in the number of confirmed cases by mean 77% was achieved in the case of mandatory wearing of N95 or equivalent masks for everyone in the hospital. The lowest effect was achieved when recommending medical or surgical masks to everyone in the hospital, resulting in a mean reduction of 17% in the number of confirmed cases.

## Discussion

During the MERS outbreak in 2015, the transmission rate in PMH was estimated to be the highest among patients, which was attributed to the spread originating from the inpatient ward. Similarly, the transmission rate induced by the superspreader was highest among

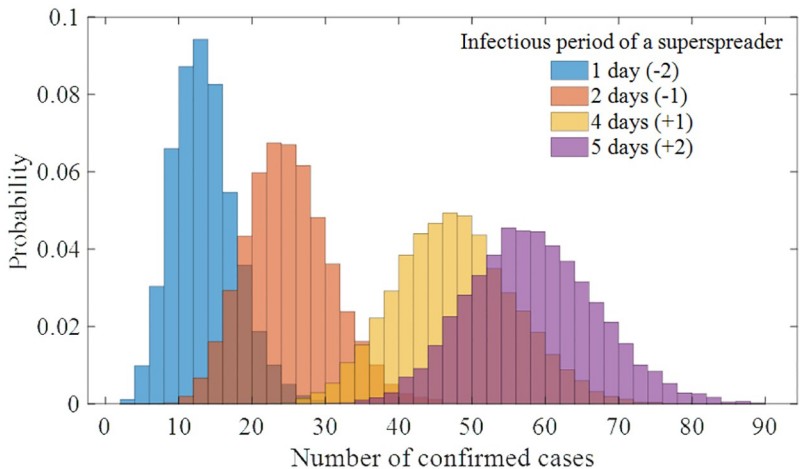

**Fig 6. Distribution of the number of confirmed cases of the model simulation runs for the varying infectious period of a superspreader (1 to 5 days).** Note that the distribution of baseline scenario simulation runs (3 days of infectious period) is not included in this figure.

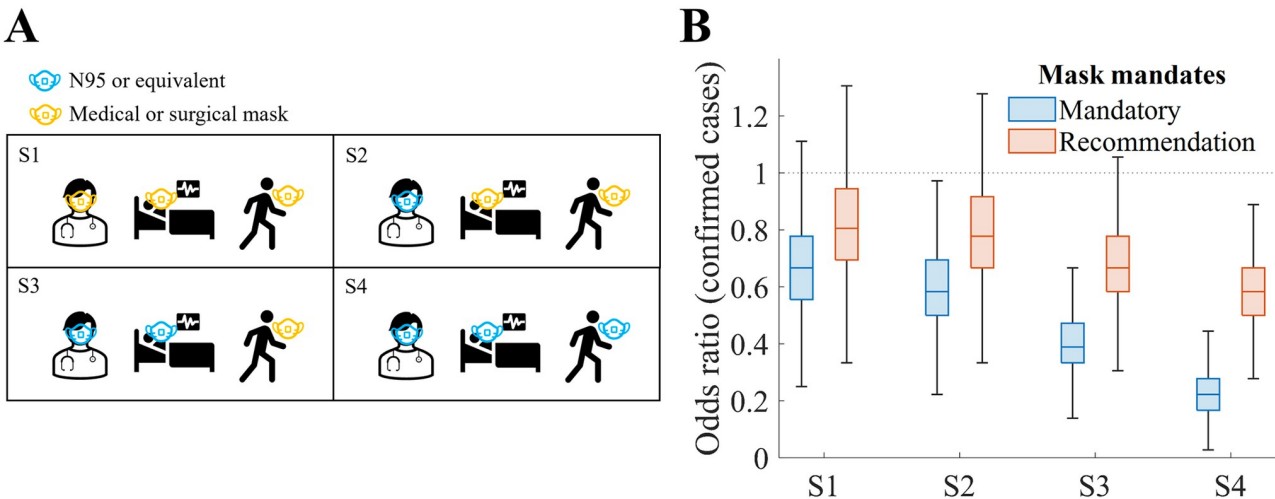

**Fig 7. Simulation results of scenarios considering different mask mandates.** (A) Description of scenario set-up, (B) odds ratio of confirmed cases. Recommendation level of intervention indicates that preventive effect of mask-wearing is reduced by half.

patients, followed by visitors and then HCWs. The estimated transmission rate directly shows the risk of the spread of infectious diseases in hospitals. Excluding the superspreader, the mean transmission rate of 0.61 can be interpreted as the possibility of an additional 0.61 infections occurring per day by a single infectious patient.

The results of the baseline scenario simulation primarily show uncertainty in the scale of the outbreak, indicating that there could have been a minimum of 23 and up to 50 confirmed cases (Fig 3). Additionally, Fig 4(C) and 4(D) show the point at which the prevalence within the hospital was expected to be greatest, which is 8 days after the onset in the baseline scenario. This indicates the time when the most infectious hosts were present. Paradoxically, this necessitates recognising the outbreak and tracking the infected individuals before the date when the spread within the hospital is most likely to occur.

Additional scenario-based analyses demonstrated the importance and potential of efforts to prevent the spread within the hospital. If the entire duration from symptom onset to isolation of infected individuals occurred within the hospital, meaning the reduction rate of the transmission period is 0% (Fig 5), there could have been more than 100 infected individuals in PMH alone. Moreover, if the period in which a single superspreader stayed in the hospital was extended to 5 days (an increase of 2 days compared to reality; Fig 6), there could have been 61% more cases.

Since the onset of the COVID-19 pandemic, mask mandates have been strongly implemented in Korea, and as of March 2024, it is mandatory to wear masks in hospitals. The results of this study show that maintaining such interventions has a significant effect on preventing the spread of infectious diseases within hospitals. The study also provides appropriate mask intervention strategies depending on the level of spread prevention goals. For example, to maintain the prevention of infectious disease spread in hospitals at more than 50%, it is essential to take protective measures for HCWs and patients (Fig 7).

The limitations of this study are as follows. (1) Detailed spaces within the hospital, such as wards and rooms, were not considered. (2) The effect of non-pharmaceutical interventions was modeled as an average. (3) Visitors entering and exiting the hospital were not considered. (4) Transmission by healthcare workers (HCWs) was assumed to be negligible. (5) We assumed a frequency-dependent transmission pattern within hospitals.

In this study, we focused on a single hospital context and did not include health-seeking behaviors—such as visiting multiple healthcare facilities, often called 'doctor shopping'—which significantly contributed to the cross-hospital spread during the 2015 MERS outbreak in the Republic of Korea [14, 39]. Addressing this and other complexities, such as detailed spatial arrangements, individual patient tracking, isolation and testing procedures, and infections involving HCWs, would allow for a more comprehensive model. Future work will expand on these elements, incorporating diverse transmission types and behaviors to better capture the dynamics of healthcare-associated outbreaks.

Additionally, our methodology could be adapted to other outbreaks in closed settings, such as those documented early in the COVID-19 pandemic in the Republic of Korea, in environments like hospitals, nursing homes, and correctional facilities [40–42]. With access to individual case data specific to each location, we believe our approach could be effectively applied to analyze similar scenarios. Furthermore, as recorded in endemic regions, future studies will highlight the transmission pathways moving from environmental sources to humans and subsequently to healthcare facilities.

## Conclusion

This study encapsulates the process of estimating transmission rates using information collected from epidemiological investigations, specifically the suspected duration of the infectious period and the disease exposure time of individuals. By using the Metropolis-Hastings algorithm for parameter estimation, we were able to present the transmission rates as distributions, illustrating the uncertainty in the transmission rate of the underlying data.

The overall methodology of this study can be applied to other outbreak situations, where it will yield different results due to spatial characteristics when applied to different countries, hospitals, or regions. This will help in ensuring preparedness and establishing intervention policies that can be tailored to the specific conditions in future outbreak situations.

## Author Contributions

**Conceptualization:** Youngsuk Ko, Eunok Jung.

**Data curation:** Youngsuk Ko.

**Formal analysis:** Youngsuk Ko, Eunok Jung.

**Funding acquisition:** Eunok Jung.

**Investigation:** Youngsuk Ko, Eunok Jung.

**Methodology:** Youngsuk Ko.

**Project administration:** Eunok Jung.

**Resources:** Youngsuk Ko.

**Software:** Youngsuk Ko.

**Supervision:** Eunok Jung.

**Validation:** Youngsuk Ko.

**Visualization:** Youngsuk Ko.

**Writing – original draft:** Youngsuk Ko.

**Writing – review & editing:** Youngsuk Ko, Eunok Jung.

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
