## [Decision Letter · Decision Letter 0]

21 Oct 2024

PONE-D-24-28130Mathematical and Statistical Approaches in Epidemiological Investigation of Hospital Infection: A Case Study of the 2015 Middle East Respiratory Syndrome Outbreak in KoreaPLOS ONE

Dear Dr. Jung,

Thank you for submitting your manuscript to PLOS ONE. After careful consideration, we feel that it has merit but does not fully meet PLOS ONE’s publication criteria as it currently stands. Therefore, we invite you to submit a revised version of the manuscript that addresses the points raised during the review process.

Your manuscript was reviewed by three experts in the field. The reviewers identified several important problems in your submission. Please review carefully the attached comments and provide point-by-point responses.

We look forward to receiving your revised manuscript.

Kind regards,

Yury E Khudyakov, PhD

Academic Editor

PLOS ONE

Journal Requirements:

1. When submitting your revision, we need you to address these additional requirements. Please ensure that your manuscript meets PLOS ONE's style requirements, including those for file naming. The PLOS ONE style templates can be found at https://journals.plos.org/plosone/s/file?id=wjVg/PLOSOne_formatting_sample_main_body.pdf and https://journals.plos.org/plosone/s/file?id=ba62/PLOSOne_formatting_sample_title_authors_affiliations.pdf 2. Please note that PLOS ONE has specific guidelines on code sharing for submissions in which author-generated code underpins the findings in the manuscript. In these cases, we expect all author-generated code to be made available without restrictions upon publication of the work. Please review our guidelines at https://journals.plos.org/plosone/s/materials-and-software-sharing#loc-sharing-code and ensure that your code is shared in a way that follows best practice and facilitates reproducibility and reuse. 3. Thank you for stating the following in the Acknowledgments Section of your manuscript: "This research was supported by the Government-wide R&D Fund Project for Infectious Disease Research (GFID), Republic of Korea (grant No. HG23C1629). This paper is supported by the Korea National Research Foundation (NRF) grant funded by the Korean government (MEST) (NRF-2021R1A2C100448711)." We note that you have provided funding information that is not currently declared in your Funding Statement. However, funding information should not appear in the Acknowledgments section or other areas of your manuscript. We will only publish funding information present in the Funding Statement section of the online submission form. Please remove any funding-related text from the manuscript and let us know how you would like to update your Funding Statement. Currently, your Funding Statement reads as follows: "This research was supported by the Government-wide R&D Fund Project for Infectious Disease Research (GFID), Republic of Korea (grant No. HG23C1629). This paper is supported by the Korea National Research Foundation (NRF) grant funded by the Korean government (MEST) (NRF-2021R1A2C100448711). The funders had no role in study design, data collection and analysis, decision to publish, or preparation of the manuscript." Please include your amended statements within your cover letter; we will change the online submission form on your behalf.

Reviewers' comments:

Reviewer's Responses to Questions

**Comments to the Author**

1. Is the manuscript technically sound, and do the data support the conclusions?

Reviewer #1: Yes

Reviewer #2: Partly

Reviewer #3: No

2. Has the statistical analysis been performed appropriately and rigorously? 

Reviewer #1: Yes

Reviewer #2: Yes

Reviewer #3: I Don't Know

3. Have the authors made all data underlying the findings in their manuscript fully available?

Reviewer #1: Yes

Reviewer #2: Yes

Reviewer #3: No

4. Is the manuscript presented in an intelligible fashion and written in standard English?

Reviewer #1: No

Reviewer #2: Yes

Reviewer #3: Yes

5. Review Comments to the Author

Reviewer #1: Dear Authors,

This is an interesting study and it enhances the existing knowledge base on modeling and parameter estimation. However, you need improve on the writing style for better content flow. Also always avoid copy-paste verbatim of content.

Reviewer #2: The study “Mathematical and statistical approaches in epidemiological investigation of hospital infection: A case study of the 2015 Middle East Respiratory Syndrome outbreak in Korea” aims to introduce a process for interpreting epidemiological investigations using mathematical and statistical methods to estimate key parameters and transmission rates.

Line 14-16: “For example, during the Coronavirus disease 2019 (COVID-19) pandemic… [8-11]”. Does the statement refer to the COVID-19 pandemic worldwide or only isolated to Korea? In the four cited references there are studies only from Korea.

Line 19-21: “By integrating epidemiological investigations… of model results”. Is this an assumption or a statement? If it is a statement, please add the reference.

Line 24-27: “As a case study, we apply this method to an outbreak at Pyeongtaek… where 36 out of 186 total cases were identified [12–14]”. According to reference 12, the 186 cases (36 death and 138 recovered cases) were reported until July 26, 2015. According to figure 2 from reference 12 "Distribution of transmission of Middle East respiratory syndrome coronavirus clusters and suspected super spreaders in South Korea (20th May to 25th November 2015)" the epidemiological analysis was carried out. What is the new information presented in your study?

Line 27-30: “The spread in Korea… than six people [15]. Of the 186 cases, it was suspected that 15 transmitted the disease to others, with five identified as superspreaders.” The "super-spread" terminology used in the text was taken from reference 15. Please add the appropriate reference.

Line 36-38: “The methodology used in this study, which relatively easily transforms the situation of hospital infections into a likelihood function, can be applied to other hospitals or infectious disease outbreaks.” Is this sentence an assumption or is it demonstrated by other articles that use this method for other infectious diseases?

Line 230-234: “The overall methodology of this study can be applied to other outbreak situations… to the specific conditions in future outbreak situations.” Considering that the present study was submitted in 2024, after the WHO declared that the COVID-19 pandemia had ended, I would have liked you to compare MERS and SARS-CoV-2 in the perspective in which the methodology used can be inter-connected.

Strong points:

- the methodology used by the authors is very good, clear and applicable;

- the mathematical approach used in this study is very useful for the prevention, analysis and establishment of biosecurity procedures.

Weak points:

- most of the epidemiological analysis was published in references 12 to 15;

- the images do not have a satisfactory clarity.

Reviewer #3: The author tried to demonstrate the mathematical and statistical approaches using historical data of the MERS outbreak in S. Korea.

It is well known that the MERS outbreak in South Korea was an explosive outbreak in healthcare settings which needs more demonstrated in the text and added in the model. The outbreak largely relied on the patient's health-seeking behaviors, so it should be reflected in their model.

Furthermore, the transmission also relies on the viral shedding in the infector and infectee; thus this factor should be reflected in the author's model.

I would strongly recommend the author read the article (PMID: 31961300) and incorporate the transmission dynamic in the author's model which primarily considers mathematics.

6. PLOS authors have the option to publish the peer review history of their article (what does this mean?). If published, this will include your full peer review and any attached files.

Reviewer #1: **Yes: **Amos Ssematimba

Reviewer #2: No

Reviewer #3: No

---

## [Author Response · Author response to Decision Letter 0]

24 Nov 2024

Reviewer #1:

This is an interesting study and it enhances the existing knowledge base on modeling and parameter estimation. However, you need improve on the writing style for better content flow. Also always avoid copy-paste verbatim of content.

Answer

We revised the paper to improve readability and avoid any potential controversy. Following are example revised parts:

Abstract: We revised the abstract section as follows for better readability and structure:

Mathematical and statistical methods are invaluable in epidemiological investigations, enhancing our understanding of disease transmission dynamics and informing effective control measures. In this study, we presented a method to estimate transmissibility using patient-level data, with application to the 2015 MERS outbreak at Pyeongtaek St. Mary’s Hospital, the Republic of Korea. We developed a stochastic model based on individual case data to derive a likelihood function for disease transmission. Through scenario-based analysis, we explored transmission dynamics, including the role of superspreaders, and investigated how mask-wearing impacted infection control within the hospital. Our findings indicated that the superspreader during the MERS outbreak had approximately 25 times higher transmissibility compared to other patients. Under scenarios of prolonged hospital transmission periods, the number of cases could potentially increase threefold. The impact of mask-wearing in the hospital was significant, with reductions in the epidemic scale ranging from 17\\% to 77\\%, depending on the type of mask and intervention intensity. This study quantifies key risk factors in hospital-based transmission, demonstrating the effectiveness of intervention measures. The methodology developed here can be readily adapted to other infectious diseases, providing valuable insights for future outbreak preparedness and response strategies.

In introduction: L2-8

An epidemiological investigation is a systematic method used to determine the cause, source, and spread of a disease within a population, usually following an unusual increase in the number of cases. This process typically involves steps like verifying the outbreak, defining and identifying additional cases, using epidemiological methods to understand the disease transmission dynamics, and collecting data pertaining to cases, their characteristics, and potential risk factors.These data are directly utilized to estimate model parameters and simulate realistic outbreak patterns. This allows for evidence-based implementation of control measures to prevent further spread of the disease and reduce the risk of future outbreaks. 

L13-17

For instance, during the COVID-19 pandemic in South Korea, detailed epidemiological information on individual cases—including estimated exposure dates, symptom onset dates, and reporting dates—was collected and analyzed using mathematical and statistical methods until the Omicron variant became dominant.

Reviewer #2: 

Comment #1

Line 14-16: “For example, during the Coronavirus disease 2019 (COVID-19) pandemic… [8-11]”. Does the statement refer to the COVID-19 pandemic worldwide or only isolated to Korea? In the four cited references there are studies only from Korea.

Answer #1

This section was included to introduce an example of large-scale collection and analysis of individual case information during the COVID-19 pandemic. We revised it as follows (L13-17): 

For instance, during the COVID-19 pandemic in South Korea, detailed epidemiological information on individual cases—including estimated exposure dates, symptom onset dates, and reporting dates—was collected and analyzed using mathematical and statistical methods until the Omicron variant became dominant.

Comment #2

Line 19-21: “By integrating epidemiological investigations… of model results”. Is this an assumption or a statement? If it is a statement, please add the reference.

Answer #2

We cited two new papers, along with four additional papers referenced in the previous sentence:

 Shim E, Choi W, Song Y. Clinical time delay distributions of COVID-19 in 2020–2022 in the Republic of Korea: inferences from a nationwide database analysis. J Clin Med. 2022;11(12):3269. 

 Ko Y, Lee J, Seo Y, Jung E. Risk of COVID-19 transmission in heterogeneous age groups and effective vaccination strategy in Korea: a mathematical modeling study. Epidemiology and Health. 2021;43.

 Ko Y, Lee J, Kim Y, Kwon D, Jung E. COVID-19 vaccine priority strategy using a heterogenous transmission model based on maximum likelihood estimation in the Republic of Korea. Int J Environ Res Public Health. 2021;18(12):6469. 

 Ko Y, Mendoza VMP, Seo Y, Lee J, Kim Y, Kwon D, et al. Quantifying the effects of non-pharmaceutical and pharmaceutical interventions against Covid-19 epidemic in the Republic of Korea: mathematical model-based Approach considering age groups and the delta variant. Math Model Nat Phenom. 2022;17:39. 

 Anderson RM, May RM. Infectious diseases of humans: dynamics and control. Oxford university press; 1991.

 Hazelbag CM, Dushoff J, Dominic EM, Mthombothi ZE, Delva W. Calibration of individual-based models to epidemiological data: A systematic review. PLoS computational biology. 2020 May 11;16(5):e1007893.

Comment #3

Line 24-27: “As a case study, we apply this method to an outbreak at Pyeongtaek… where 36 out of 186 total cases were identified [12–14]”. According to reference 12, the 186 cases (36 death and 138 recovered cases) were reported until July 26, 2015. According to figure 2 from reference 12 "Distribution of transmission of Middle East respiratory syndrome coronavirus clusters and suspected super spreaders in South Korea (20th May to 25th November 2015)" the epidemiological analysis was carried out. What is the new information presented in your study?

Answer #3

In this study, we presented insights into estimating transmission rates within the context of a specific outbreak at Pyeongtaek, utilizing general mathematical and statistical methods tailored for epidemiological investigations. We addressed the heterogeneity of hosts in a hospital environment, categorizing individuals into superspreaders, healthcare workers, patients, and visitors. This allowed us to develop a model that accurately reflects the dynamics of the outbreak and simulates realistic transmission scenarios. We also demonstrated that this method could be easily applied elsewhere. To enhance readability, we moved the section discussing “estimating transmission rate” forward and revised the sentences accordingly (L32-42).

To investigate transmission rates in detail, we categorized individuals into superspreaders, healthcare workers (HCWs), patients, and visitors based on actual epidemiological investigations. The spread in Korea was largely due to superspreaders, patients who caused secondary infections in more than six people. Of the 186 cases, it was suspected that 15 transmitted the disease to others, with five identified as superspreaders.\\\\

Simultaneously with the estimation of the transmission rate, we developed a model that reflects these investigations and simulates a realistic outbreak.A scenario-based analysis was introduced to quantify risk factors such as the infectious period of infected individuals, including superspreaders, and the effectiveness of mask mandates within the hospital.  

Comment #4

Line 27-30: “The spread in Korea… than six people [15]. Of the 186 cases, it was suspected that 15 transmitted the disease to others, with five identified as superspreaders.” The "super-spread" terminology used in the text was taken from reference 15. Please add the appropriate reference.

Answer #4

After reviewing the content, we added the following references. These studies include definitions of superspreaders, associated risks, and cases from Korea:

[Additional references]

 Lloyd-Smith JO, Schreiber SJ, Kopp PE, Getz WM. Superspreading and the effect of individual variation on disease emergence. Nature. 2005 Nov 17;438(7066):355-9.\\\\

 Cho SY, Kang JM, Ha YE, Park GE, Lee JY, Ko JH, Lee JY, Kim JM, Kang CI, Jo IJ, Ryu JG. MERS-CoV outbreak following a single patient exposure in an emergency room in South Korea: an epidemiological outbreak study. The Lancet. 2016 Sep 3;388(10048):994-1001.\\\\

 Park JE, Jung S, Kim A, Park JE. MERS transmission and risk factors: a systematic review. BMC public health. 2018 Dec;18:1-5.

Comment #5

Line 36-38: “The methodology used in this study, which relatively easily transforms the situation of hospital infections into a likelihood function, can be applied to other hospitals or infectious disease outbreaks.” Is this sentence an assumption or is it demonstrated by other articles that use this method for other infectious diseases?

Answer #5

In response to the comment, we revised it for clarity. This revision emphasizes that the methodology is not merely an assumption but is supported by its successful application in various contexts, thereby demonstrating its relevance and effectiveness in estimating transmission rates in different infectious disease scenarios. The updated version (L42-49) states:

The methodology employed in this study transforms the dynamics of hospital infections into a likelihood function, allowing for its application to other hospitals or infectious disease outbreaks. By focusing on the estimation of transmission rates among heterogeneous hosts in certain spaces like hospitals, we expect these findings to significantly enhance both reactive and preventive measures. This approach, which has been successfully utilized in various contexts, including human-to-human and animal-to-animal infectious diseases, underscores the potential for improving infection control strategies

[Additional references]

 Keeling MJ, Woolhouse ME, Shaw DJ, Matthews L, Chase-Topping M, Haydon DT, Cornell SJ, Kappey J, Wilesmith J, Grenfell BT. Dynamics of the 2001 UK foot and mouth epidemic: stochastic dispersal in a heterogeneous landscape. Science. 2001 Oct 26;294(5543):813-7.

 Boender GJ, Hagenaars TJ, Bouma A, Nodelijk G, Elbers AR, de Jong MC, Van Boven M. Risk maps for the spread of highly pathogenic avian influenza in poultry. PLoS computational biology. 2007 Apr;3(4):e71.

Comment #6

Line 230-234: “The overall methodology of this study can be applied to other outbreak situations… to the specific conditions in future outbreak situations.” Considering that the present study was submitted in 2024, after the WHO declared that the COVID-19 pandemia had ended, I would have liked you to compare MERS and SARS-CoV-2 in the perspective in which the methodology used can be inter-connected.

Answer #6

In the Republic of Korea, during the early stages of the COVID-19 pandemic, there were several reports of outbreaks in closed environments such as hospitals, nursing homes, and correctional facilities. If we can obtain individual case data specific to each location, it seems feasible to apply this methodology. Taking this comment into account, I will aim to conduct comparative studies in the future when we have processed data categorized by space. We added following sentence in discussion section (L253-259)

Additionally, our methodology could be adapted to other outbreaks in closed settings, such as those documented early in the COVID-19 pandemic in the Republic of Korea, in environments like hospitals, nursing homes, and correctional facilities. With access to individual case data specific to each location, we believe our approach could be effectively applied to analyze similar scenarios. Furthermore, as recorded in endemic regions, future studies will highlight the transmission pathways moving from environmental sources to humans and subsequently to healthcare facilities.

[Added references]

 Lee HA, Ahn MH, Byun S, Lee HK, Kweon YS, Chung S, Shin YW, Lee KU. How COVID-19 affected healthcare workers in the hospital locked down due to early COVID-19 cases in Korea. Journal of Korean Medical Science. 2021 Dec 6;36(47).

 Giri S, Chenn LM, Romero-Ortuno R. Nursing homes during the COVID-19 pandemic: a scoping review of challenges and responses. European Geriatric Medicine. 2021 Dec;12(6):1127-36.

 Lee HY, Park YJ, Yu M, Park H, Lee JJ, Choi J, Park HS, Kim JY, Moon JY, Lee SE. Accuracy of rapid antigen screening tests for SARS-CoV-2 infection at correctional facilities in Korea: March-May 2022. Infection \\& Chemotherapy. 2023 Dec;55(4):460.

Reviewer #3: 

Comment #1

It is well known that the MERS outbreak in South Korea was an explosive outbreak in healthcare settings which needs more demonstrated in the text and added in the model. The outbreak largely relied on the patient's health-seeking behaviors, so it should be reflected in their model.

Answer #1

In the actual 2015 MERS outbreak, health-seeking behaviors (also referred to as “doctor shopping”) contributed to the spread across multiple hospitals. In our study, however, we limited the model to a single hospital (Pyeongtaek St. Mary's Hospital) and did not consider this behavior. In reality, case #14, responsible for further spread, was initially infected at Pyeongtaek and subsequently admitted to a larger hospital (Samsung Medical Center), where subsequent transmissions occurred. We believe that health-seeking behaviors could be effectively addressed in an advanced model that considers both hospitals and the community. To address this comment, we have added the following statement in the Discussion section (L239-259):

In this study, we focused on a single hospital context and did not include health-seeking behaviors—such as visiting multiple healthcare facilities, often called 'doctor shopping'—which significantly contributed to the cross-hospital spread during the 2015 MERS outbreak in the Republic of Korea. Addressing this and other complexities, such as detailed spatial arrangements, individual patient tracking, isolation and testing procedures, and infections involving HCWs, would allow for a more comprehensive model. Future work will expand on these elements, incorporating diverse transmission types and behaviors to better capture the dynamics of healthcare-associated outbreaks.

Additionally, our methodology could be adapted to other outbreaks in closed settings, such as those documented early in the COVID-19 pandemic in the Republic of Korea, in environments like hospitals, nursing homes, and correctional facilities. With access to individual case data specific to each location, we believe our approach could be effectively applied to analyze similar scenarios. Furthermore, as recorded in endemic regions, future studies will highlight the transmission pathways moving from environmental sources to humans and subsequently to healthcare facilities.

[Additional references]

 Kim SW, Park JW, Jung HD, Yang JS, Park YS, Lee C, Kim KM, Lee KJ, Kwon D, Hur YJ, Choi B. Risk factors for transmission of Middle East respiratory syndrome coronavirus infection during the 2015 outbreak in South Korea. Clinical Infectious Diseases. 2017 Mar 1;64(5):551-7.  

Comment #2

The transmission also relies on the viral shedding in the infector and infectee; thus this factor should be reflected in the author's model.

Answer #2

In our study, the transmission rate incorporates both the number of contacts per unit time and the probability of infection following those contacts. This naturally includes viral shedding. We have accounted for host heterogeneity, which can vary based on factors such as distance from the infector and contact frequency, even if the shedding remains constant among infectors. To reflect this aspect, we have revised the text in section L88-93 as follows:

Infectious hosts transmit the disease through viral shedding. We assumed that there is heterogeneity in the successful transmission based on contact frequency, duration, and distance between the infectee and infector, considering the types of hosts (HCWs, patients, and visitors) within the hospital setting.

Comment #3

I would strongly recommend the author read the

---

## [Decision Letter · Decision Letter 1]

22 Dec 2024

Mathematical and Statistical Approaches in Epidemiological Investigation of Hospital Infection: A Case Study of the 2015 Middle East Respiratory Syndrome Outbreak in Korea

PONE-D-24-28130R1

Dear Dr. Jung,

We’re pleased to inform you that your manuscript has been judged scientifically suitable for publication and will be formally accepted for publication once it meets all outstanding technical requirements.

Kind regards,

Yury E Khudyakov, PhD

Academic Editor

PLOS ONE

Additional Editor Comments (optional):

Reviewers' comments:

Reviewer's Responses to Questions

**Comments to the Author**

1. If the authors have adequately addressed your comments raised in a previous round of review and you feel that this manuscript is now acceptable for publication, you may indicate that here to bypass the “Comments to the Author” section, enter your conflict of interest statement in the “Confidential to Editor” section, and submit your "Accept" recommendation.

Reviewer #1: All comments have been addressed

Reviewer #2: (No Response)

2. Is the manuscript technically sound, and do the data support the conclusions?

Reviewer #1: (No Response)

Reviewer #2: Yes

3. Has the statistical analysis been performed appropriately and rigorously? 

Reviewer #1: (No Response)

Reviewer #2: Yes

4. Have the authors made all data underlying the findings in their manuscript fully available?

Reviewer #1: (No Response)

Reviewer #2: Yes

5. Is the manuscript presented in an intelligible fashion and written in standard English?

Reviewer #1: (No Response)

Reviewer #2: Yes

6. Review Comments to the Author

Reviewer #1: (No Response)

Reviewer #2: (No Response)

7. PLOS authors have the option to publish the peer review history of their article (what does this mean?). If published, this will include your full peer review and any attached files.

Reviewer #1: **Yes: **Amos Ssematimba

Reviewer #2: No

---

## [Editor Report · Acceptance letter]

13 Jan 2025

PONE-D-24-28130R1 

PLOS ONE

Dear Dr. Jung, 

I'm pleased to inform you that your manuscript has been deemed suitable for publication in PLOS ONE. Congratulations! Your manuscript is now being handed over to our production team.

Kind regards, 

on behalf of

Dr. Yury E Khudyakov 

Academic Editor

PLOS ONE